# Study of an Anti-Doping Education Program in Spanish Sports Sciences Students

**DOI:** 10.3390/ijerph192316324

**Published:** 2022-12-06

**Authors:** Carlos García-Martí, Jonathan Ospina-Betancurt, Eva Asensio-Castañeda, José L. Chamorro

**Affiliations:** 1Faculty of Sport Sciences, Universidad Europea de Madrid, 28670 Villaviciosa de Odón, Spain; 2Hum-878 Research Team, Health Research Centre, Department of Psychology, University of Almería, 04120 Almería, Spain

**Keywords:** doping, education, Spain, anti-doping intervention, PASS

## Abstract

Doping continues to be one of the biggest risks to the credibility of elite sports, and its practice remains widespread among athletes despite improved controls. Athletes’ support personnel could be key to preventing doping behavior. In this sense, anti-doping education for this population appears as a possible strategy to reduce doping behaviors in elite sport, but these programs must be evaluated and designed based on scientific evidence. The aim of this research is to explore the impact of an anti-doping education program about substances perceived efficacy, ill-health short- and long-term effects, and the morality of doping substance use in Spanish sports sciences students. Method: A total of 145 students of Physical Activity and Sport Sciences (PASS) from different Spanish universities who took an online anti-doping educational course of the Spanish Anti-Doping Commission (CELAD) answered a questionnaire on their perceptions about doping before, after, and four months later. Results: The results show that the course reduced students’ ignorance about the effects of substances on performance and health and increased their moral judgment and feelings against doping. Discussion: The results are in line with previous research that showed that the moral stance against doping can be improved through educational programs. Conclusion: Online educational interventions can be effective in reducing doping behavior, so their future implementation among ASP can be an effective strategy to reduce doping behavior.

## 1. Introduction

Doping is undoubtedly one of the greatest challenges facing the sports world in the coming decades, as recognized by sports organizations [1], since it endangers athletes’ health, jeopardizes sports’ integrity, and negatively impacts elite sports’ legitimacy. The creation of the World Anti-Doping Agency in 1999 has allowed for greater legal international harmonization, increased funding for anti-doping policy, and more effective detection systems [2]. However, doping remains difficult to tackle and widespread in elite sports: in the review by de Hon et al. [3], the authors stated that doping ranged between 14 and 39% of adult elite athletes, while using a “randomized response technique” to guarantee anonymity, Ulrich et al. [4] found a prevalence among athletes at two international events between 44 and 57%.

Doping itself is a complex phenomenon resulting from the interaction of a set of institutional settings and psychosocial factors [5]. In their review, Backhouse et al. [6] highlight, firstly, that doping takes place within a complex social context, including institutional and economic constraints, social norms, and coaching culture. In this context, both situational temptations together with certain psychological factors can foster or limit the probability of doping.

In particular, several psychosocial factors have been proven to influence doping behavior and doping susceptibility. Noutmanis et al. [7] found in a meta-analysis that perceived social norms, use of supplements, and positive attitudes towards doping were positively correlated with doping intentions, while self-efficacy and morality correlated negatively.

Evidence in the few last years has also confirmed this link between morality and doping likelihood [8,9,10]. The idea is that individuals develop a moral identity—a will to act according to what they believe is right and wrong—as part of their broader social identity, and that moral identity includes a set of beliefs, attitudes, and behaviors [11].

Once their moral identity has been defined and integrated, individuals try to act according to their own moral standards to be true to themselves and maintain a positive self-image. To act morally, they need moral reasoning [12], but also self-regulatory mechanisms and the perceived ability to act morally, following the self-determination theory [13].

Even though individuals all have a moral identity, how important it is for them to act accordingly may vary, as will its content: different people will consider different attitudes and behaviors as important. In the case of sports, fair play is considered one of its core values, and therefore sportspeople who consider fair play to be morally important should be more likely to reject doping as an unfair practice. This has been the case with Lithuanian university athletes [14], British team sports athletes [8], and British, Danish, and Greek footballers [15].

If a strong moral identity should prevent doping offenses, the ability to downplay the moral consequences of the actions should have the opposite effect. Moral disengagement [16] has been defined as precisely that: to minimize negative emotions arising from immoral behavior using different psychological strategies such as stating that everybody does it—diffusion of responsibility—or that the individual is following orders—displacement [17].

Moral disengagement must deal with moral feelings of guilt, shame, and embarrassment triggered by self-defined immoral behaviors. As a result, this anticipated guilt should prevent moral disengagement and lower the likelihood of doping, as seen in team sports athletes [8,9,15] and other antisocial sporting behaviors [17,18].

Together with morality, positive attitudes towards doping are one of the most predicting variables of doping likelihood [7], including doping’s perceived benefits and negative outcomes such as ill-health effects, since believing doping increases performance and carries little to no harm fosters doping behavior [19]. Serious health risks have been shown to deter doping behavior [20], although it seems difficult for athletes to think about the long-term effects of doping abuse and focus more on perceived short-term benefits [21].

In terms of doping effects and anti-doping policy, athletes have shown important knowledge gaps [6]. For example, 41.1% of a sample of Turkish athletes considered caffeine to be on the prohibited substances list [22], and Australian athletes only scored 32.2% correct on substances on that list [23].

Taking into consideration the high doping abuse rates [3], the influence of psychosocial factors [7], and the lack of knowledge [6], anti-doping education is clearly needed. To date, there is still a lack of efficient educational interventions, evidence showing moderate impact, and a lack of long-term effect, probably due to design pitfalls such as being too short [24], a mismatch between objectives, or a floor effect, and participants already having strong anti-doping beliefs prior to the intervention [25]. However, in recent years, some successful interventions educating on the negative health effects of doping and ethical values have been developed [26,27].

Following the need for evidence-based education, National Anti-Doping Organizations (NADOs) have started including education in their policies, although not systematically. Gatterer et al. reviewed NADOs’ programs and found that only 58% of NADOs had educational programs, mostly with a knowledge-based approach, but there was still a lack of value-based, social skills, and affective approaches [28]. Additionally, WADA has developed its own education strategy, which includes the online program ADEL and the creation of an official standard to harmonize anti-doping education effective as of 1 January 2021 [29].

Together with athletes, research and education efforts have also been devoted lately to Athlete Support Personnel (ASP)—coaches, physicians, trainers, or family members—since they are part of the athletes’ moral climate, which has been determinant in their behavior [30]. Research shows that they share similar gaps in knowledge [31], especially coaches, who lack resources [32] and may develop a passive line of action since they prioritize performance and do not consider anti-doping education a core part of their job [33].

Physical Activity and Sports Sciences (PASS) students include some of the future generations of ASP, so anti-doping education during their education could be a fruitful strategy to ensure that this collective is ready and willing to participate in anti-doping efforts [34]. However, there has been limited research on this group, with mixed results. Vangrunderbeek and Tolleneer [34] found that students in the period from 1998 to 2006 were shifting their opinion from an 85% zero tolerance at the beginning of the period to less than 50% at the end, which could mean that PASS were moving towards a more tolerant position. A few years later, Takahashi et al. [35] found 20% support for doping among Japanese students. In Spain, results from two studies both showed strong opposition to doping among students [36], with only 5.75% of respondents in one survey supporting doping legalization [37]. If knowledge about doping attitudes among PASS students is scarce, education is apparently absent. To our knowledge, although there are education programs for medical students [38], there is not a program for PASS students, which makes this study especially appropriate.

For these reasons, the primary goal of this study is to investigate the impact of an anti-doping education program on the perceived efficacy of substances, the short- and long-term effects of ill-health, and the morality of doping substance use in Spanish PASS students.

## 2. Materials and Methods

PASS students must be of legal age—over 18 years—to be eligible. The study passed the university ethics committee, and all participants signed an informed consent form after receiving information about the study and their rights regarding data privacy and free participation and dropout.

Participants: The Spanish Anti-Doping Agency enrolled 16 sports sciences schools, which offered the program to their students enrolled in the Physical Activity and Sports Sciences grade. Participants read and signed an informed consent, and their enrollment was voluntary. Confidentiality was guaranteed, and no personal information was collected through the questionnaires. A total of 145 students were enrolled and finished the course. Students’ mean age was 21.4 years (17–41; SD 4.5), and 73.1% were men. They all answered the questionnaire before starting the course (PRE), while 54 (37.2%) answered it right after completion (POST1) and 46 (31.7%) did it at the four-month follow-up (POST2).

Educational intervention: the Spanish national anti-doping organization, CELAD, designed and implemented an educational program specially designed for PASS students called *Vive Sin Trampas* (Live Without Cheating).

The program was developed by CELAD initially for high-school PE teachers and adapted later for PASS students. It includes information on anti-doping regulations, doping history, doping consequences, sports values, sociopsychological doping variables, and doping prevention (see Table 1 for detailed information on the program). It was theoretically based on the Teaching Personal and Social Responsibility Model (TPSRM) by Donald Hellison [39], which has been widely used with young athletes and students before [40] to foster self-efficacy, respect, and social skills.

The program took place between September 2020 and January 2021. Due to COVID-19 pandemic government policies, the entire program was delivered online. It started with a 1 h, live, introductory online seminar explaining the main features and milestones of the program. Then, students completed a 25 h online course, including activities and debates corrected and guided by an online instructor from CELAD. Finally, a 1 h online live seminar was delivered specifically focused on how to implement the Teaching Personal and Social Responsibility Model on doping.

Students finishing the course received an official diploma, and their schools recognized its completion as 1 ECTS (European Credit Transfer System) for voluntary, out-of-program education valid for their academic curriculum.

Instruments: A Spanish short version of the WADA-designed questionnaire for evaluating educational programs was used [41] to measure the impact of the *Vive Sin Trampas* program. The Spanish version of this questionnaire has been validated in a sample of Spanish athletes showing good reliability [42]. This instrument is based on the Sport Drug Control Model developed by Donovan et al. [19], which states that doping behavior depends on the athlete’s attitudes and intentions, which are themselves determined by the benefits and threats appraisal and personal morality. All these individual and context variables are also impacted by the sport’s socio-economic context and the broad socio-cultural context [43].

In our study, since participants were PASS students, not all dimensions of the model were included since they were not considered to be at-risk athletes but future ASP. Therefore, and based on the aim of the present study, only questions about doping substances perceived efficacy (e.g., ‘If you were to use the following substances, how likely is it that these substances would improve your performance in your sport?’), ill-health short-term (e.g., ‘How much harm to your health do you think would be caused by using each of the following substances for a short time say up to two months?’), long-term effects (e.g., ‘How much harm to your health do you think would be caused by using each of the following substances regularly?’), moral judgment (e.g., ‘Regardless of whether you believe Performance Enhancing Substances or Methods (PESM) should be banned or allowed, which of the following statements best describes your own personal feelings about deliberately using banned PESM?’), and moral emotions (e.g., ‘If you were caught using banned performance enhancing substances or methods, to what extent would you experience the following feelings: ashamed, embarrassed, guilty?’) regarding doping were selected to the data analyses. Questions about substances’ efficacy used a 5-point Likert scale, while questions about substances’ health effects used a 4-point Likert scale. In both cases, a ‘don’t know’ option was available. As for questions on morality, the question about moral stance on doping used a 3-point Likert scale, while the questions about moral feelings used a 5-point Likert scale. Questions about doping behavior and doping controls’ experiences were not included since participants were not elite athletes.

The instrument was delivered online through the Lime Survey platform, retaining no personal data and assuring the participants’ anonymity. No time limit for completing the questionnaire was included.

Data analyses: the data were analyzed using the IBM SPSS version 27.0 software package (Armour, NY, USA). To test the effect of the educational program on knowledge of PESM use, frequency analyses were carried out between yes/no knowledge through the three times of measurement (i.e., PRE, POST1, and POST2). In addition, a chi-squared test was performed to explore the differences among times of measurement.

To explore the effects of the educational program on moral judgments about PES use, the Kruskal–Wallis test and a Games-Howell post-hoc analysis were performed due to the non-parametric nature of the data. To explore the effect of the program on moral emotions, an analysis of variance (ANOVA) was carried out, and variables with a significant difference were subjected to post hoc analysis using Bonferroni correction. The effect sizes were calculated by *η*^2^ with a Mann–Whitney U Test and an ANOVA using G-Power software. A value of *p* < 0.05 was considered statistically significant.

## 3. Results

### 3.1. Effects of the Educational Program on Knowledge of PES Use

Figure 1, Figure 2 and Figure 3 show the evolution of ignorance rates about PESM performance enhancement efficiency and short- and long-term ill-health effects, i.e., the percentage of respondents who answered ‘don’t know’ when asked about it. As can be seen, this percentage dropped drastically after the intervention for all substances, both in POST1 and POST2. Table 2 shows that χ^2^ test results were statistically significant for all differences found between PRE and POST1 and POST2 measures. In all cases, knowledge about PES performance-enhancing, short- and long-term health effects increased significantly (*p* < 0.05) for all substances between PRE and POST1, and this difference remained similar four months later (POST2).

### 3.2. Effects of the Educational Program on Morality

First, students showed a great moral rejection of doping before the intervention (82.1% rejected doping in any situation), but their rejection increased after the course (POST1 = 96.3%) and remained higher 4 months later (POST2 = 91.3%). A Kruskal–Wallis test shows a significant positive effect of the educational program on moral judgment on PES use. H(_gl_) = 8.46(_2_), *p* = 0.015. *Post-hoc* analysis using the Games-Howell test revealed that the POST1 group scored higher on considering PES use always morally wrong (Mdn = 1) under all circumstances compared to the pre-intervention group (Mdn = 2, *p* = 0.002) IC 95% [0.95, 2.25], with a moderate effect size of 1 − β = 0.86 and d = 0.05 (1 − β = 0.86).

Cronbach’s alpha (α = 0.91) revealed that the moral emotions scale was extremely reliable. ANOVA tests also identified the effect of the program on participants’ moral emotions—shame, embarrassment, and guilt—F(_2241_) = 3.103, *p* = 0.047, with a significant low size (*η*^2^ = 0.16). Post-hoc analysis using Bonferroni also showed that participants on POST2 scored significantly higher on moral emotions about PES use (M = 4.70, SD = 0.62, *p* = 0.03) IC 95% [4.51, 4.88] than the PRE group (M = 4.36, SD = 1.06) IC 95% [4.19, 4.54].

## 4. Discussion

Developing effective interventions to reduce doping behavior has been acknowledged as a key instrument to reduce current high doping rates [4]. Until now, interventions lacking purposeful design and sufficient length have lacked significant impact [24]. Additionally, there is a lack of interventions on ASP, and previous studies have shown that PASS students lack adequate knowledge about doping substances, regulations, sanctions, and prevention strategies [44,45]. This education intervention proved to be effective in two ways: it increased perceived knowledge about doping substances and increased moral rejection of doping. As stated by Aguilar-Navarro et al. [44], PASS students may not need to be doping legal regulations experts, but it is essential that they understand its negative physical, psychological, and social consequences on the athletes.

### 4.1. Perceived Knowledge

As mentioned earlier and stated in the Sport Drug Control Model, perceived efficacy and health effects are key in triggering doping use. If ASP must act as doping deterrents, they must have a strong belief that doping is an unsafe behavior. A significant number of students before the intervention ignored substances’ efficiency (ranging from 17.9% in the case of human growth hormone to 54.5% for designer steroids such as tetrahydrogestrinone), substances’ short-term ill-health effects (ranging from 26.6% for steroids to 44.1% for beta blockers), and substances’ long-term effects (ranging from 24.1% for steroids to 39.3% for diuretics). This lack of knowledge matches previous research and confirms the educational needs of the ASP to become reliable athletes’ advisors. As stated by Blank [45], athletes and their ASP lack enough knowledge about the consequences of doping, mentioning only sport sanctions (35.8% of athletes and 35.9% of the ASP), while criminal sanctions were only mentioned by 2.1% of the ASP and no athletes. They also failed to mention other types of consequences, such as future health issues or social and family impacts.

The positive results are even more consistent if we consider that the effects remained in place 4 months after the intervention. Long-term effects are the most challenging dimension of educational intervention since other programs have found their effects dissipate after some time [24,46]. Typically, this loss is attributed to the inadequate length or intensity of the programs, which would mean that a 25 h intervention is enough to provide an effect, despite its online administration. Importantly, this intervention also included standard doping information coupled with doping sociopsychological dimensions such as risk factors and trigger situations, personality traits, decision-making, and counseling. As proven in other interventions [25,45], to achieve impactful results, it is important to include not only doping knowledge but also sports values, psychological variables, and a discussion about the moral decision to turn to doping.

### 4.2. Moral Judgment

The intervention also changed students’ moral judgments on doping. As previously stated [7], morality has been shown to be one of the variables that best correlate with doping behavior, and therefore anti-doping education must develop a strong moral opposition among athletes and ASP to be effective. Previous research by Sukys et al. [14] has also shown that the attitudes of athletes and the ASP towards doping are more positive compared to those of non-athletes. In this sense, students showed a high moral rejection of doping before taking the course (82.1%), but after the intervention, the number of students who did not justify doping under any circumstances increased substantially (96.3%) and remained higher 4 months later (91.3%). Again, these are positive results that show a path to effective educational interventions.

These positive outcomes refer to an understudied group [36], sports sciences students, which we consider may be a key group both in elite and recreational settings since they will develop their careers in the sports world. According to Chirico et al. [47], athletes typically seek to indulge in doping to (re)gain personal significance, developing moral disengagement and dissociation strategies to justify their conduct. ASPs trained in anti-doping skills should be better equipped to identify possible at-risk individuals and prevent doping abuse. In order to do so, ASP needs doping education not only about substances’ health effects and legal framework but also about sociopsychological factors influencing doping behavior.

## 5. Conclusions

This study provides evidence that long, intensive online interventions produce positive effects on attitudes, knowledge, and moral standing toward doping behavior among sports sciences students. Anti-doping education is required for PASS students and athletes’ support personnel to carry out their professional duties effectively. Offering these students programs that include information about doping offenses, doping effects, and sociopsychological variables may be an effective way to increase anti-doping attitudes and reduce doping behavior.

## 6. Limitations

This study has some limitations that must be taken into account when interpreting its results. This was a pre-post study and as such has limitations regarding sample selection: students were all voluntarily enrolled, so there must be a bias since all students had a prior interest in doping and maybe were already more prone to reject doping than their non-participating peers. Additionally, this was not a randomized controlled trial, and as such, there was no random allocation preventing selection bias. There was also no control group, so we cannot know if changes would have occurred naturally over time. There was also a limited retention rate (31.7% at POST2), although this was in line with other anti-doping education efforts [48]. However, future interventions should develop methods to increase retention.

## Figures and Tables

**Figure 1 ijerph-19-16324-f001:**
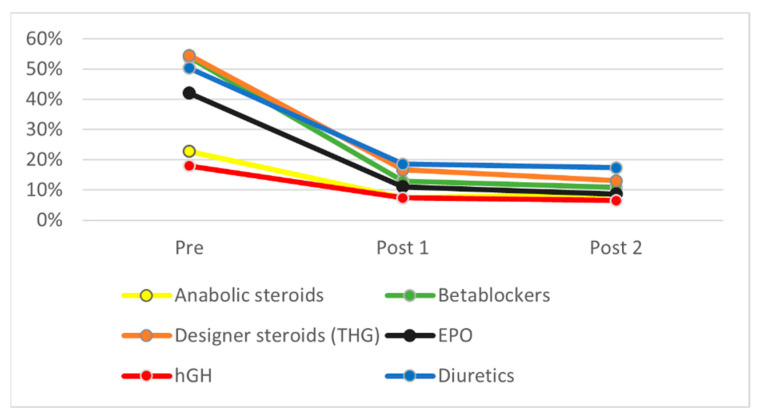
General ignorance about PES use on performance.

**Figure 2 ijerph-19-16324-f002:**
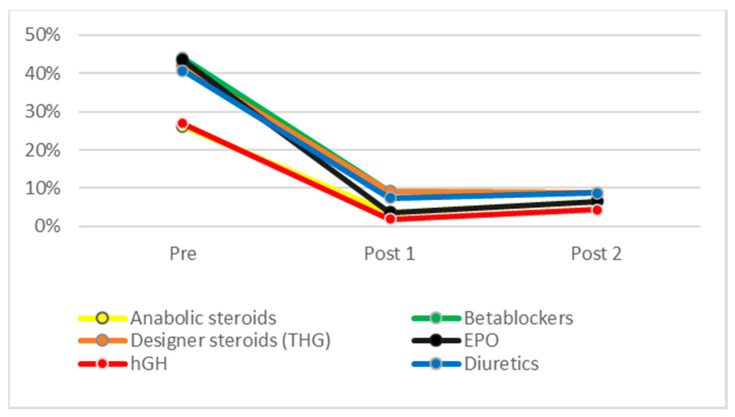
General ignorance about PES use on health in the short-term.

**Figure 3 ijerph-19-16324-f003:**
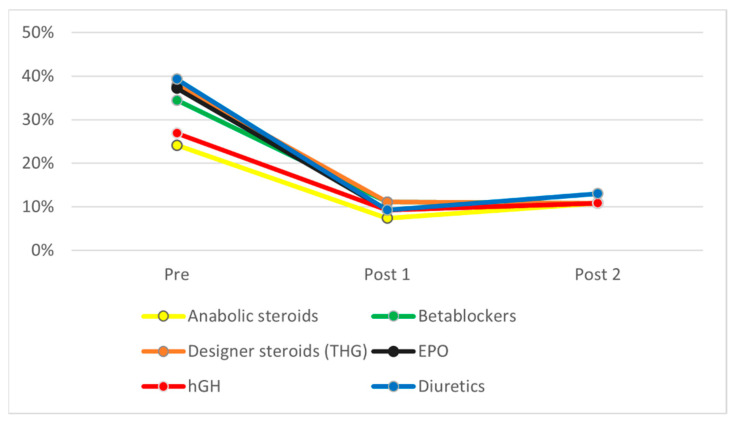
General ignorance about PES use on health in the long-term.

**Table 1 ijerph-19-16324-t001:** Participants and characteristics of a *Vive Sin Trampas* (Live Without Cheating) anti-doping education program in Spanish sports sciences students.

Phase	Aims	Content
Phase 1: 1 h, introductory online seminar	Introduction to the course	Sports values and their relationship with fair play and doping prevention.
Phase 2: 25 h online course		
Unit 1: Doping	Teaching the main features of the anti-doping regulation	History and definition of doping. Regulations and a list of prohibited substances and methods in sports. Consequences of doping. Prevention of unintentional doping.
Unit 2: Doping psychological variables	Train students on the psychological factors as well as the periods of greatest risk and vulnerability.	Education in values. Assertiveness and the decision-making process. Risk and protective factors. The different roles in prevention: sources of influence. Points of intervention in situations of doping temptation. Assessment of vulnerability and attitudes.
Phase 3: 1 h live online seminar	To train students into using the TPSRM in doping scenarios	The 4 key competencies of the sports educator are to promote a culture of learning and fair play in the group. The 5 levels of the Personal and Social Responsibility Program. The 9 methodological strategies of the sports educator promote positive development and healthy habits. Structure and organization of a session to integrate personal and social responsibility and promote positive development through sport.

**Table 2 ijerph-19-16324-t002:** Differences in PES use knowledge between three moments of educational intervention.

	PRE				POST1				POST2						
	Yes		No		Yes		No		Yes		No				
	N	%	N	%	N	%	N	%	N	%	N	%	χ^2^	p	µ^2^
Knowledge PES use—performance effects															
Anabolic steroids	112	77.2	33	22.8	50	92.6	4	7.4	42	91.3	4	8.7	9,28	0.01	0.195
Beta blockers	67	46.2	78	53.8	47	87	7	13	41	89.1	5	10.9	44.53	0.001	0.426
Designer steroids (THG)	66	45.5	79	54.5	45	83.3	9	16.7	40	87	6	13	39.15	0.001	0.400
EPO	84	57.9	61	42.1	48	88.9	6	11.1	42	91.3	4	8.7	29.64	0.001	0.348
hGH	119	82.1	26	17.9	50	92.6	4	7.4	43	93.5	3	6.5	6.08	0.048	0.158
Diuretics	72	49.7	73	50.3	44	81.5	10	18.5	38	82.6	8	17.4	26.54	0.001	0.329
Knowledge PES use—long-term effects															
Anabolic steroids	110	75.9	35	24.1	50	92.6	4	7.4	41	89.1	5	10.9	9.40	0.009	0.196
Beta blockers	95	65.5	50	34.5	48	88.9	6	11.1	41	89.1	5	10.9	17.45	0.001	0.267
Designer steroids (THG)	90	61.1	55	37.9	48	88.9	6	11.1	41	89.1	5	10.9	21.81	0.001	0.298
EPO	91	62.8	54	37.2	49	90.7	5	9.3	41	89.1	5	10.9	22.79	0.001	0.305
hGH	106	73.1	39	26.9	49	90.7	5	9.3	41	89.1	5	10.9	10.60	0.005	0.108
Diuretics	88	60.7	57	39.3	49	90.7	5	9.3	40	87	6	13	23.83	0.001	0.312
Knowledge PES use—short-term effects															
Anabolic steroids	107	73.8	38	26.6	52	96.3	2	3.7	43	93.5	3	6.5	18.53	0.001	0.275
Beta blockers	81	55.9	64	44.1	49	90.7	5	9.3	42	91.3	4	8.7	34.94	0.001	0.378
Designer steroids (THG)	84	57.9	61	42.1	49	90.7	5	9.3	42	91.3	4	8.7	31.72	0.001	0.360
EPO	82	56.6	63	43.4	52	96.3	2	3.7	43	93.5	3	6.5	43.73	0.001	0.422
hGH	106	73.1	39	26.9	53	98.1	1	1.9	44	95.7	2	4.3	23.90	0.001	0.312
Diuretics	86	59.3	59	40.7	50	92.6	4	7.4	42	91.3	4	8.7	31.85	0.001	0.361

Notes: χ^2^ = chi-squared; µ^2^ = eta squared; *p* = 0.05 IC 95%.

## Data Availability

The data presented in this study are available on request from the corresponding author. Due to legal issues, the data are not publicly available.

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
