# Peer review of "Study of an Anti-Doping Education Program in Spanish Sports Sciences Students"

_ijerph, 2022, doi:10.3390/ijerph192316324_

Round 1

Reviewer 1 Report

i am grateful for the opportunity to review your manuscript.

i offer the following for your consideration:

- please do a global search of the body text of your document for the acronyms 'AST' and 'ASP', to ensure consistency of use, such that your eventual chosen acronym aligns with 'Athlete Support Personnel'.

- do consider adding a paragraph break such that the final sentence of your Introduction stands alone; the merit of doing so would be that the aim of your study would be more explicitly communicated to your eventual readers.

Author Response

As authors of the manuscript, we want to thank the three reviewers for their positive comments on our manuscript and its potential contribution. We also want to thank all of you for the comments and recommendations aimed at improving the quality of the manuscript. In the following letter we specify all the changes done following your advice.

Reviewer 2 Report

The abbreviations ASP and TPSRM have no explanation anywhere in the text.

The discussion in both chapters should be expanded

Author Response

(The authors gave the same response as above.)

Reviewer 3 Report

What were the criteria for selecting the students?

What were the criteria for excluding some questions from the presented tool?

How did you come to select a number of questions? Why weren't other questions selected? what was the scientific argument?

What was the response time to these questions?

Did you have in your attention Cronbach’s alpha indicator to  measure internal consistency for your items?

What is the value of the pretest as long as it was followed by the providing of information about the subject?

Why didn't all students answer post test 1? Were those who answered post test 2 the same students or those who did not answer post test 1?

What is the presentation of the aspects resulting from the statistical analysis?

Are percentage charts advanced statistics?

How was the confidentiality aspect of the answers respected?

How can a course change a behavior that students did not have before?

How many elite athletes were part of the surveyed group?

How is moral judgment made? How did you realize that this changed in 25 hours? Is that so?

I think that the approach to the subject is forced. The entire article must be rewritten from a correct perspective.

We cannot say that a course for students will change the approach and behavior related to doping of high performance athletes.

Author Response

(The authors gave the same response as above.)

Round 2

Reviewer 3 Report

The problem of this article is one of essence, of approach.

I do not agree that an online course can bring changes of a "moral" and "attitude" nature, especially when the students are not involved in the phenomenon of high-performance sports.

The approach of this article is forced and I want to bring together the educational activity with the idea of high performance sport.

The changes brought are not of a nature to bring improvements in the nature of the approach.

The article has a "scientific guise" but no foundation in reality.

Author Response

Dear reviewer,

Thanks for your time in reviewing our manuscript and for your valuable feedback.

Since unfortunately you don’t consider it suitable for publication, we have made no further modifications to the draft and we’ll wait for the editors’ decision.

Best regards,

The authors
